# Development of a Method of Measuring β-D-Glucan and Its Use in Preemptive Therapy for Invasive Fungal Infections

**DOI:** 10.3390/ijms22179265

**Published:** 2021-08-27

**Authors:** Minoru Yoshida

**Affiliations:** Fourth Department of Internal Medicine, Teikyo University School of Medicine, Mizonokuchi Hospital, Kawasaki 213-8507, Japan; myoshida@med.teikyo-u.ac.jp

**Keywords:** β-D-glucan, horseshoe crab, galactomannan, invasive fungal infection, *Aspergillus*, *Candida*, empirical therapy, pre-emptive therapy

## Abstract

Invasive fungal infections (IFIs) are serious infections that develop in conjunction with neutropenia after chemotherapy for acute leukemia or with hematopoietic stem cell transplantation. Conventionally, empirical antifungal therapy was recommended to treat IFIs for patient safety despite a lack of evidence of fungal infections. However, many studies have indicated that antifungals were not necessary for over half of patients, and several detriments of empirical therapy were noted, e.g., antifungals caused adverse reactions, an increase in drug-resistant fungi was a possibility, and medical costs soared. β-D-glucan (BDG) is a component of clinically important fungi such as *Aspergillus* and *Candida*. The G-test was developed in Japan as a way to measure BDG in serum using a coagulation factor from the blood of the horseshoe crab. Pre-emptive antifungal therapy based upon serodiagnosis with a BDG or galactomannan assay and CT imaging has been introduced. With pre-emptive antifungal therapy, the prognosis is equivalent to that with empirical therapy, and the dose of the antifungal has been successfully reduced. Measurement of BDG has been adopted widely as a method of diagnosing IFIs and is listed in the key guidelines for fungal infections and febrile neutropenia.

## 1. Introduction

Invasive fungal infections (IFIs) are serious infections that develop in conjunction with neutropenia after chemotherapy for acute leukemia or with hematopoietic stem cell transplantation (HSCT). *Candida* species and *Aspergillus* species are the major fungi that cause IFIs, with candidemia and invasive pulmonary aspergillosis (IPA) being the predominant mycoses. Both conditions lack characteristic clinical manifestations and there is no method of diagnosing either early, so definitive diagnosis of those conditions was often delayed and their subsequent therapeutic outcomes were poor. Conventionally, empirical therapy with an antifungal is recommended to treat IFIs. Empirical antifungal therapy (EAT) is a strategy of administering an antifungal automatically in cases of refractory febrile neutropenia (FN). This therapy is based on research into empirical therapy with an antifungal conducted by the European Organization for Research and Treatment of Cancer (EORTC) over 30 years ago. A trial compared 68 patients who were administered amphotericin-B (Am-B) when FN was unresponsive to a broad-spectrum antimicrobial administered for 4 d and 64 control patients who received the same antibiotics [1]. The fever was lowered in 47 (69%) of the 68 patients receiving Am-B versus 34 (53%) of the control patients. One of the patients receiving Am-B developed a fungal infection, versus 6 of the control patients; 4 of the patients in the control group died, while none of the patients receiving Am-B did. Because this strategy allowed aggressive chemotherapy and HSCT and improved patient prognosis, the FN guidelines of the Infectious Diseases Society of America (IDSA) recommended empirical therapy with an antifungal [2]. However, the study indicated that an antifungal was not necessary for over half of the control group, and the study also stated that an IFI was present in at most about 33% of the patients for whom EAT was indicated according to the IDSA’s guidelines. At the time, however, there was no method for diagnosing fungal infections other than blood culture, and there was no method of prophylaxis using azole antifungals. In addition, the only antifungal that could be administered via intravenous infusion was Am-B, which is no longer used due to adverse reactions such as fever and renal dysfunction. As various new antifungals have subsequently been developed, several detriments of EAT have been noted, e.g., patients often received an antifungal unnecessarily, antifungals caused adverse reactions, an increase in drug-resistant fungi was a possibility, and medical costs soared. In response, pre-emptive antifungal therapy, where treatment is initiated if fungal cell components or genes are detected or based on results on CT images of the lungs, was put forth [3]. Pre-emptive antifungal therapy is also referred to as presumptive therapy and diagnostic-driven therapy.

(1→3)-β-D-glucan (BDG) is a component of the fungal cell wall and is not found in bacteria or human cells, so it is a suitable substance to screen for IFIs. The G-test was developed in Japan as a way to measure BDG in serum using a coagulation factor from the blood of the horseshoe crab [4,5,6]. In Japan, the G-test was covered by health insurance for serodiagnosis of fungal infections in 1995, ahead of the US and Europe, and it is widely used in hematology, where IFIs are often encountered [7]. The current work explains the significance of BDG in the management of IFIs.

## 2. Development of Systems to Measure BDG and Their Clinical Use

The clinical features of IFIs caused by *Candida* and *Aspergillus* differ substantially. *Candida* causes various infections in organs throughout the body, but it mainly causes a bloodstream infection, and pulmonary infections due to *Candida* are extremely rare. In contrast, an infection with *Aspergillus* takes various forms depending on the host’s immune status, such as IPA, chronic pulmonary aspergillosis (CPA), and simple pulmonary aspergilloma (SPA), though a blood culture is usually negative. IPA is known to be a typical type of IFI that occurs after chemotherapy or HSCT for acute leukemia, and it is known to have a very poor prognosis. In contrast, types of IFIs in patients with a hematologic disorder can be differentiated relatively easily based on clinical manifestations, a blood culture, and CT images. In clinical settings, whether an IFI is present or not needs to be diagnosed early based on serodiagnosis, and antifungal treatment needs to be started promptly rather than taking time to identifying the species of fungus. BDG is a component of the cell wall of many fungi and cannot be used to differentiate the species of fungus, but it can serve as a clinically useful method of screening for IFIs [7].

The G-test was developed by Obayashi et al. in 1992 [4]. Factor G has been found to be a constituent that reacts with BDG from fungi, so the principle of the G-test is to use the coagulation cascade mediated by this factor in a *Limulus amebocyte* lysate test, which is conventionally used to detect endotoxin. The principle of measuring BDG using blood of the Chinese horseshoe crab, *Tachypleus tridentatus*, is shown in Figure 1. Multicenter clinical trials conducted primarily in the fields of hematology and respiratory medicine have yielded excellent results, i.e., positive G-test results for 37 of 41 cases of IFIs with a sensitivity of 90% and a specificity of 100% [5]. Of the 41 cases, 29 were candidiasis, 8 were aspergillosis, 2 were trichosporonosis, and 2 were some other mycosis. The Fungitec G-test MK, which is a kinetic chromogenic assay, has been widely used to measure BDG [6]. The Wako β-glucan test [8], which is a kinetic turbidimetric assay, has also been used to measure BDG. Both assays are covered by health insurance in Japan (the former in 1995 and the latter in 1997). Sensitivity is crucial to early diagnosis, and the former has superior sensitivity to the latter.

In cases of candidemia, BDG measurement correlates extremely closely with a blood culture, and BDG measurement may be positive prior to a blood culture depending on the patient [9]. Moreover, BDG measurement may be positive in cases of systemic candidiasis even if a blood culture fails to detect *Candida*, and BDG measurement is highly regarded for its clinical usefulness. In cases of confirmed IPA, positivity for BDG was 82.4% according to the G-test, 84.6% according to the Fungitec G-test MK, and 72.7% according to the Wako β-glucan test, respectively [10]. However, since IPA is a condition with a poor prognosis, it needs to be diagnosed in its early stages, and a comprehensive diagnosis should be made based on measurement of BDG, detection of the *Aspergillus* GM antigen, detection of *Aspergillus* genes with PCR, and chest CT [11,12]. An unfortunate fact is that PCR testing is not covered by health insurance in Japan, hampering its widespread use. In light of this background, BDG has helped to diagnose IFIs and to help facilitate early treatment, thus improving prognosis as well [10,13,14,15].

BDG is not present in the cell wall of mucormycetes, so it tests negative in cases of mucormycosis, and it is present in small amounts in *Cryptococcus*, so it tests negative except in cases of a bloodstream infection [4,5,7]. Over the past two decades, cases of relatively rare fungal infections with a *Fusarium*, *Trichosporon*, or other spp. have increasingly been reported in the area of hematologic disorders. An early study already reported that BDG was positive in those cases as well [16]. *Tachypleus tridentatus* is difficult to obtain in Japan right now, so the Fungitec G-Test MKII “Nissui,” which uses the Atlantic horseshoe crab, *Limulus polyphemus*, is used [17].

In the US, Associates of Cape Cod (ACC) developed Fungitell, a measurement system similar to the Fungitec G-Test MK that uses *L. polyphemus*. Odabasi et al. reported that Fungitell had a sensitivity of 100% in detecting an infection in a group of 20 subjects with 16 proven or 4 probable IFIs [18]. Sixteen of the subjects had fungemia, which was caused by *Candida* in 11, *Trichosporon* in 3, *Fusarium* in 1, and *Aspergillus* in 1. Four of the 20 subjects had probable fungal pneumonia, which was caused by *Aspergillus* in 3 and *Fusarium* in 1. Fungitell was approved as a diagnostic kit for fungal infections by the FDA in 2004. This has allowed the measurement of BDG in the US and Europe, leading to the inclusion of BDG measurement in the diagnostic guidelines cited later.

## 3. Assessment of BDG in Guidelines for Fungal Infections and FN

### 3.1. EORTC/MSG Definitions of IFIs

In 2002, diagnostic criteria for IFIs were issued by the EORTC/Mycoses Study Group (MSG of the National Institute of Allergy and Infectious Diseases) based on the accuracy of diagnosis. Cases were classified as a “proven” IFI, a “probable” IFI, a “possible” IFI, or “other” (an unexplained fever not meeting criteria for a fungal infection) [19]. Proven IFI required only that a fungus be detected by histological analysis or culture of a specimen of tissue taken from a site of disease. Probable and possible IFIs hinged on 3 elements namely, a host factor that identified the patients at risk, clinical signs and symptoms consistent with the disease entity, and mycological evidence that encompassed culture and microscopic analysis, but also indirect tests, such as *Aspergillus* GM antigen detection.

Probable IFI required at least 1 host factor criterion and 1 microbiological criterion and 1 major (or 2 minor) clinical criteria from an abnormal site consistent with infection. Host factors include (1) neutropenia (<500 neutrophils/mm^3^ for >10 d), (2) persistent fever for >96 h refractory to appropriate broad-spectrum antibacterial treatment in high-risk patients, (3) body temperature either >38 or <36 °C and any of the following predisposing conditions: prolonged neutropenia (>10 d) in previous 60 d, recent or current use of significant immunosuppressive agents in previous 30 d, proven or probable IFI during a previous episode of neutropenia, or coexistence of symptomatic AIDS, (4) signs and symptoms indicating graft-versus-host disease, particularly severe (grade ≥ 2) or chronic extensive disease, and (5) prolonged (>3 weeks) use of corticosteroids in previous 60 d.

Microbiological criteria require (1) a positive result of a culture for mold (*Aspergillus, Fusarium, Zygomycetes*, etc.) or *Cryptococcus neoformans* or an endemic fungal pathogen from sputum or bronchoalveolar lavage fluid samples, (2) a positive result of a culture or findings from a cytologic/direct microscopic evaluation for mold from a sinus aspirate specimen, (3) positive findings from a cytologic/direct microscopic evaluation of mold or *Cryptococcus* spp. from sputum or bronchoalveolar lavage fluid samples, (4) a positive result for the *Aspergillus* antigen in specimens of bronchoalveolar lavage fluid, CSF, or ≥2 blood samples, (5) a positive result for the cryptococcal antigen in a blood sample, (6) positive findings from a cytologic or direct microscopic examination for fungal elements in sterile body fluid samples (e.g., *Cryptococcus* species in CSF), and (7) a positive result of a blood culture for *Candida* spp., etc.

Clinical criteria for a lower respiratory tract infection require any of the following new infiltrates on CT imaging: a halo sign, an air-crescent sign, or a cavity within an area of consolidation (major), or symptoms of a lower respiratory tract infection (cough, chest pain, hemoptysis, or dyspnea); physical findings from a pleural rub; any new infiltrate not fulfilling the major criterion; or pleural effusion (minor). In the case of a CNS infection, major criteria require radiological evidence suggesting a CNS infection (e.g., mastoiditis or other parameningeal foci, extradural empyema, or an intraparenchymal brain or spinal cord mass lesion), and minor criteria are focal neurological symptoms and signs (including focal seizures, hemiparesis, and cranial nerve palsies); mental changes; meningeal irritation findings; abnormalities in CSF biochemistry and cell count (provided that CSF is negative for other pathogens according to a culture or microscopy and negative for malignant cells).

A possible IFI requires at least 1 host factor criterion and 1 microbiological or 1 major (or 2 minor) clinical criterion from an abnormal site consistent with infection. The EORTC/MSG diagnostic criteria are most widely used in the US and Europe.

Since their revision in 2008, the criteria have included BDG as an indirect test for invasive fungal disease (IFD) other than cryptococcosis and zygomycosis [20]. However, the criterion for a “possible” IFD was defined more strictly to include only those cases with the appropriate host factors and with sufficient clinical evidence consistent with IFD such as CT but for which there was no mycological support, compared to a positive finding according to either clinical criteria or mycological criteria in the original version.

In the 2020 revision, BDG plays an even bigger role in the serodiagnosis of IFDs in appropriate clinical settings [21]. This includes patients with hematologic malignancies with and without neutropenia, neutropenia following HSCT, and certain patients in the ICU who are at higher risk (>10%) for invasive candidiasis as a result of gastrointestinal surgery with recurrent anastomotic leaks, perforations of the upper gastrointestinal tract, or necrotizing pancreatitis when there is clinical suspicion of infection.

### 3.2. IDSA Guidelines for Aspergillosis and Candidiasis

BDG was mentioned in the IDSA’s practice guidelines for aspergillosis published in 2008, and the assay used to detect it was described as a variation of the *Tachypleus* or *Limulus* assay used to detect endotoxin [22]. The results of multicenter studies of the G-test, which uses reagents derived from *Tachypleus tridentatus* [4,5], and Fungitell, which uses reagents derived from *Limulus polyphemus*, are cited [18,23]. At this point in time, the database for this assay was limited to patients with hematological disorders, and the guidelines stated that further studies needed to be conducted in other populations. Data from use of Fungitell were compiled in the 2016 version of the IDSA’s guidelines [24]. According to those updated guidelines, comparative studies have shown that the Fungitell assay can be slightly more sensitive than GM for IA but is limited by its poor specificity, while others have found that Fungitell is not as helpful for IA. However, another study in a large cancer center that compared GM and BDG prospectively over a 3-year period in 82 patients, each for 12 weeks, found that the BDG was more sensitive than the GM assays for detection of IA and other mold infections in patients with hematological malignancy [25]. In either event, a comprehensive judgment needs to be made not with measurement of BDG alone but also with detection of the GM antigen and PCR, if possible.

Non-culture diagnostic tests for invasive candidiasis were rarely covered in the IDSA’s 2009 practice guidelines for candidiasis [26], but the 2016 edition featured data from use of Fungitell to detect BDG [27]. The updated guidelines state: “True-positive results are not specific for invasive candidiasis, but rather suggest the possibility of an IFI. For this reason, among patient populations that are also at risk for invasive mold infections, such as hematopoietic cell transplant recipients, BDG offers a theoretical advantage over more narrow assays for candidiasis. BDG detection can identify cases of invasive candidiasis days to weeks prior to positive blood cultures and shorten the time to initiation of antifungal therapy. In meta-analyses of BDG studies, the pooled sensitivity and specificity for diagnosing invasive candidiasis were 75–80% and 80%, respectively. Due to the uncertainty or low sensitivity of blood culture in the diagnosis of invasive candidiasis, non-culture diagnostic tests for blood culture-negative invasive candidiasis have also been discussed, with PCR and BDG being the focus of much attention.” This position is very close to and in line with the sentiments of Japanese hematologists.

### 3.3. Japanese IFI Guidelines

Guidelines for diagnosis and treatment of IFIs published by a mycosis forum are the most widely used guidelines on IFIs in Japan. The first edition of Japanese domestic guidelines for management of deep-seated mycosis were originally published in 2003, followed by a second edition published in 2007 [28,29]. The second edition was revised and subsequently published in 2014 as the Japanese Domestic Guidelines for Management of Deep-seated Mycosis 2014 [30]. The English version of the Executive Summary was published in 2016 [31]. The Japanese guidelines feature flowcharts for diagnosis and treatment for each clinical department as well as added explanations. The following 10 clinical departments are listed and arranged so that readers can refer to the portion of interest: (1) hematology, (2) internal medicine and respiratory medicine, (3) surgery, emergency medicine, and intensive care, (4) organ transplantation, (5) pediatrics, (6) ophthalmology, (7) otolaryngology, (8) HIV, (9) imported mycosis, and (10) infection control. The level of diagnosis is also divided into suspected cases, clinically diagnosed cases, and proven cases. These diagnostic categories correspond to possible IFI, probable IFI, and proven IFI as proposed in the EORTC/MSG criteria. A sample flowchart for diagnosis in hematology is shown in Figure 2. Although the flow chart is rather complicated, it shows steps in the search for IFIs in patients with FN for which broad-spectrum antibiotics are not effective, which is useful for hematologists. BDG has been at the center of serodiagnosis since the first version of the Japanese guidelines was published in 2003, and its position in the serodiagnosis of IFIs has not changed. BDG is one of the mycological factors for candidiasis in hematology as well as in surgery and emergency medicine. Other *Candida* antigen tests including the mannan antigen remain unverified and are not recommended. In routine practice, BDG is measured when following up on IFIs, and BDG may be used to determine therapeutic efficacy or, in some cases, to indicate when to halt an agent.

### 3.4. Guidelines for FN in USA and Japan

Since Fungitell was not available until 2004, BDG was not mentioned in the IDSA’s FN guidelines in 1990, 1997, or 2002 [2,32,33], but it was finally mentioned in the 2011 revision as a method of serodiagnosis that could be used in pre-emptive antifungal therapy [34]. In the event that an infection is unresponsive to antimicrobials, the Japanese FN guidelines indicate that BDG should be measured to test for IFIs along with a GM antigen test and a CT scan of the chest [7,35,36,37].

## 4. Empirical Therapy and Pre-Emptive Therapy for IFIs

Therapy with an antifungal has been actually divided into targeted therapy after a definitive diagnosis and empirical therapy for suspected cases. The latter is a treatment where, if FN is unresponsive to a broad-spectrum antimicrobial after 4 or 5 d, an antifungal is added automatically despite a lack of evidence of fungal infections. In Japan, however, evidence of fungal infections was sought in those instances, so BDG has been measured as appropriate [13,14].

The new echinocandin antifungal micafungin (MCFG) came onto the market in 2002 [38], and Japan’s first guidelines for diagnosis and treatment of IFIs were published the following year [28]. Since then, many new antifungal agents, such as voriconazole and liposomal amphotericin B, have been approved for use in Japan, and research into treatments in this area has progressed rapidly. Patients with hematologic disorders who were treated with MCFG were assembled post-marketing. An initial study from April 2003 to March 2005 assembled 277 subjects [39], and a second study from April 2005 to September 2006 assembled 541 subjects [40]. Combined, the 2 studies had 618 subjects in whom efficacy could be evaluated; 474 received chemotherapy and 144 underwent HSCT. A “proven” fungal infection was noted in 14 subjects (2.3%), a “probable” fungal infection was noted in 69 (11.2%), a “possible” fungal infection was noted in 211 (34.1%), and a persistent fever unresponsive to antimicrobials was noted in 324 (52.4%). In the real world, over half of patients received an antifungal with no evidence of a fungal infection. The second study measured BDG in 360 subjects, and results were positive in 47 (13.1%). The EORTC/MSG’s diagnostic criteria at the time deemed a positive result on diagnostic imaging or a serodiagnosis to be a “possible” IFI [19], so the diagnosis was upgraded from an unexplained fever. Eighty-two subjects were tested for the GM antigen, and only 2 (2.4%) were positive; both were positive for BDG as well. A decrease in BDG was noted in 24 of 33 cases (72.7%) where BDG was measured after treatment with MCFG. Approximately 10% of patients who received EAT with MCFG in the study were diagnosed with an IFI based on BDG data, but conversely, a significant number of other patients were found to have been overtreated.

Pre-emptive antifungal therapy combining serodiagnosis or genetic testing using PCR and CT images has been introduced over the past two decades [41,42,43]. Fung et al. conducted meta-analysis comparing empiric versus pre-emptive antifungal strategies in FN among adult patients with a hematologic malignancy [44]. Data were collected in 9 studies from 1998 to 2009, and BDG was used for diagnosis in only one study by Oshima et al. [45]. IFIs are slightly more apparent with pre-emptive antifungal therapy than with EAT due to the delay of starting antifungals in the former, but the prognosis is equivalent and the dose of the antifungal has been successfully reduced. Recently, a combination of the D-Index (an indicator of the severity and duration of neutropenia), serodiagnosis, and diagnostic imaging in the CEDMIC trial successfully reduced the MCFG dose from 60.2 to 32.5% while ensuring efficacy and safety [46]. The trial recommended weekly BDG and GM testing, chest X-rays, and CT scans as needed. Two hundred and sixty subjects in that trial developed persistent FN; 17 (6.5%) were BDG-positive and 23 (8.8%) were GM-positive [47].

Ceesay et al. used the revised EORTC/MSG criteria to examine the incidence of IFDs in patients undergoing HSCT or receiving aggressive chemotherapy who were predicted to have neutropenia persisting for 10 d or longer [48]. Of 209 patients, 44 had a “proven” or “probable” IFD; 22 were positive for GM and 34 were positive for BDG. They stated that performing both tests was particularly useful.

The Supportive Care Committee of the Japan Adult Leukemia Study Group (JALSG) conducted surveys on supportive care during the treatment of acute leukemia at participating facilities [49,50,51,52]. Table 1 shows whether empirical therapy or pre-emptive therapy was chosen as the treatment strategy for IFIs and the testing performed in the event of the latter. BDG and GM were covered by health insurance in Japan starting in 1995. A look at the usage of BDG and the GM antigen indicates that BDG measurement was included in routine practice in over 90% of cases when the initial survey was conducted in 2001. Use of GM was initially limited because the Pastorex *Aspergillus* test had a low sensitivity, but the Platelia *Aspergillus* test had an improved sensitivity. A proposal was made to lower the cut-off level from 1.5 to 0.5 in Dec, 2006, and then use of the latter test increased [53]. Testing twice a week was recommended in the US and Europe, but testing in Japan had to be paid for by health insurance, so it was often performed once a week. PCR is not covered by health insurance, so it is seldom performed even today. If a fever is unresponsive to antimicrobials after 4 d, conventional EAT automatically adds an antifungal. Many Japanese hematologists now recognize that EAT is largely a waste. In fact, they treat patients with careful attention to the use of diagnostic imaging and serodiagnosis. In such instances, measurement of BDG is an essential test as a marker of IFIs.

## 5. Discussion

BDG currently plays an important role in the diagnosis of IFIs along with a blood culture, CT, and testing for the GM antigen. BDG increases with *Candida* and *Aspergillus*, which are 2 major genera of fungi that are clinically important. BDG is a key method of testing to determine whether or not IFIs are caused by those fungi. EAT is a treatment strategy that is widely used today. However, EAT has also been examined as part of antifungal stewardship in comparison to pre-emptive antifungal therapy to avoid overtreatment. In Japan, leukemia treatment and HSCT are provided in a more stringent protective environment than in the US or Europe, so IFIs will presumably be less frequent than in the US or Europe. The IDSA’s FN guidelines estimate the incidence of IFIs at the start of EAT to be around 33% at most [2,32]. An analysis in a multicenter study of MCFG in Japan indicated that positivity for BDG + GM was 13.1% in the early 2000 s, and more recently it has been 15.3% [40,46,47]. In the event of prophylaxis with an azole agent or providing treatment in a protective environment, EAT should not be uniformly used to treat FN that is unresponsive to antimicrobials. Even if the sensitivity of a diagnostic test is taken into account, a treatment strategy must be based on serology that suggests some type of IFI.

Mucormycetes are the second filamentous fungi most likely to cause an invasive pulmonary infection next to *Aspergillus* [54]. BDG is not a component in the cell wall of mucormycetes, so it cannot help to diagnose that condition. There is no useful serodiagnosis for mucormycosis, but conversely mucormycosis is more likely in cases of pneumonia that is unresponsive to antimicrobials in persistent FN where GM and BDG are negative. BDG is also negative in cases of pulmonary cryptococcosis, but that condition is less prevalent in the area of hematologic disorders, so it will not pose a major problem clinically. BDG is also positive in cases of rare fungal infections, which have increased over the past two decades [5,16,23]. Given these facts, BDG is an extremely useful way to screen for IFIs. Although not touched upon in the current work, the BDG level is also high in cases of *Pneumocystis jiroveci* pneumonia (PCP) [55]. Measurement of BDG is useful in the management of PCP in patients with HIV and it can also help to determine therapeutic efficacy.

Studies have cited several precautions when using BDG. A considerable number of studies have reported false-positives in particular [5,7,14,18,23]. These are mainly iatrogenic patient contamination through the use of medical devices containing BDG and parenterally-delivered materials as well as translocation of intestinal luminal BDG due to mucosal barrier injury [56]. Ultimately, serodiagnosis should be used as a diagnostic aid and should be performed in a patient population with a prevalence of IFIs. Many false-positives can be differentiated if properly evaluated by an experienced clinician. The same is also true for GM [57]. Measurement of BDG has a 25-year history; it was developed in Japan and has been adopted widely as a method of diagnosing IFIs. When BDG is used as one index at the start of antifungal therapy, unnecessary use of antifungals can be avoided while maintaining safety and efficacy.

## Figures and Tables

**Figure 1 ijms-22-09265-f001:**
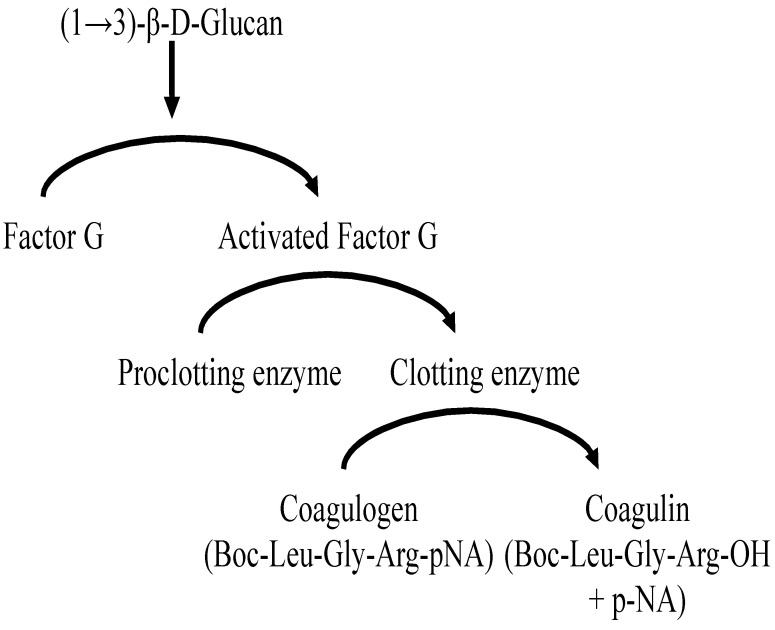
Principle of G test.

**Figure 2 ijms-22-09265-f002:**
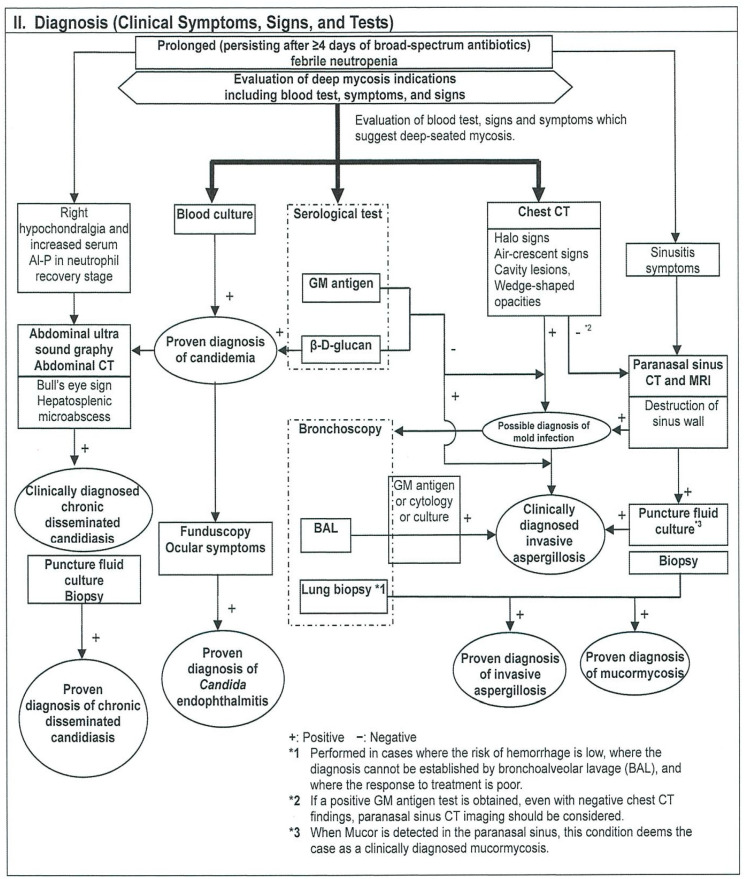
Flowchart for hematological disease.

**Table 1 ijms-22-09265-t001:** Treatment strategy of the invasive fungal infection in JALSG.

Year	2001	2007	2013	2019
Empiric	(-)	54%	59%	51%
Pre-emptive	(-)	42%	40%	43%
Biomarker				
BDG	98%	99% (55%)	99% (65%)	98% (59%)
GM	43%	75% (28%)	96% (40%)	93% (40%)
Target	(-)	0%	0%	1%

Notes: JALSG: Japan Adult Leukemia Study Group. (-): No question. (%): More than once a week.

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
