# Peer review of "Development of a Method of Measuring β-D-Glucan and Its Use in Preemptive Therapy for Invasive Fungal Infections"

_ijms, 2021, doi:10.3390/ijms22179265_

Round 1
Reviewer 1 Report
The manuscript offers a concise, clear historical review of diagnostic and treatment strategies for invasive fungal infections along with the limitations of conventional, empirical antifungal therapy. Also, the authors provide a solid rationale for pre-emptive antifungal therapy based on multiple diagnostic tools, such as CT imaging plus serodiagnosis along with the b-D-glucan (BDG) or the galactomannan assay. Diagnostic strategies and clinical criteria for highly relevant mycoses caused by a diversity of fungal etiologic agents, including fungi particularly belonging to the genera Candida and Aspergillus, and also Fusarium, Trichosporon, Cryptococcus, and Pneumocystis, along with the mucormycetes, are discussed. Figure 1 (Principle of G Test) and Figure 2 (Flowchart for hematological disease) offer informative and clear visuals for the readers. The following list includes suggestions for the author’s consideration:
Line 16: Consider changing therapy combining serodiagnosis …
To therapy based upon serodiagnosis ….
Line 27: Consider changing Candida and Aspergillus
To Candida species and Aspergillus species
Line 28: Change main types to predominant mycoses
Line 36: Omit also
Line 42: Consider revising sentence to: Because this strategy allowed aggressive chemotherapy and HSCT and it improved patient prognosis, the FN guidelines of the Infectious Diseases Society of America (IDSA) recommended empirical therapy with an antifungal [2].
Lines 85-88: Consider revising sentence to:
Multicenter clinical trials conducted primarily in the fields of hematology and respiratory medicine have yielded excellent results, i.e. positive G-test results for 37 of 41 cases of IFIs with a sensitivity of 90% and a specificity of 100% [5].
Lines 89-90: Change Some other fungal infection To some other mycosis
Lines 89-90: Change Some other fungal infection To some other mycosis
Line 316: Revise. the only method to a key method This suggestion is made because other methods in the diagnostic lab, such as histopathologic detection of Candida and Aspergillus
Check format for titles of articles in References to assure that they are consistent regarding use of capitalization for the first letter of only the first word of each article’s title or of words throughout titles.
Lines 377 and 378. Review and omit. LSEP. From Ref #6
Line 403. Add period After the word Years
Line 406: Italicize Fusarium Trichosporon Saccharomyces Acremonium
Line 409: Change to: “Nissui.” with a space Italicize Med. Mycol. J.
Line 413: Change Syndrome To syndrome
Lines 432. Review and omit. LSEP. From Ref #23
Line 439. Add period after J and after Clin and after Microbiol
And Line 507 (Ref #48) and check for consistency throughout References
Line 471: Review and omit. LSEP. From Ref #38
Author Response
Thank you for your careful peer review and proofreading to the smallest detail.
All the corrections were made as instructed without any objection.
I also tried to check the style of the cited references as much as possible.

Reviewer 2 Report
The manuscript of Minoru Yoshida titled “Development of a method of measuring β-D-glucan and its use in preemptive therapy for invasive fungal infections” gives an extensive overview on the different detection and treatment protocols for fungal infections. This review is well written in sound English, it is concise and easy to follow. I suggest its acceptance in IJMS after a minor revision.
Minor comments:
The author states that Candida infections are not as dangerous as the ones caused by other fungi. One of the greatest problems in hospitals is the appearance of multidrug resistant Candida Auris. I think it should be mentioned in the Introduction part.
Figure 1 should be placed closer to its first mention in the text. This is also valid for the other figures and tables. This is, however more an editing than a scientific issue.
In line 257 the author mentions: “Many new antifungals have been developed since…” A few references on these new antifungals should be inserted here, for example the original molecule presented in: (Nagy et al, Molecules 25(4):903, DOI:10.3390/molecules25040903) and a few other molecules from the recent years.
Author Response
Thank you for your kind suggestions.
This article does not argue that Candida infection is less dangerous than other fungal infections. We do not have data on β-glucan in Candida auris, and we believe that there is no need to cite it for the purpose of this manuscript.
I agree with the positions of the figures and tables. I would like to move to P2 line56 (or P3 top), P5 line 232, and P7 line 298, respectively.
The wording of the sentence on line 257 was inaccurate. It is accurate that not many drugs were developed, but many drugs such as voriconazole and liposomal amphotericin B were approved for use in Japan after the launch of micafungin. I changed the text that way. Therefore, I do not think that it is necessary to cite a paper on drug development.
